# High Enzymatic Recovery and Purification of Xylooligosaccharides from Empty Fruit Bunch via Nanofiltration

**Hans Wijaya [1,2], Kengo Sasaki [3], Prihardi Kahar [3]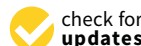, Nanik Rahmani [2], Euis Hermiati [4], Yopi Yopi [5], Chiaki Ogino [1,*], Bambang Prasetya [5] and Akihiko Kondo [1,3]**

[1] Department of Chemical Science and Engineering, Graduate School of Engineering, Kobe University, 1-1 Rokkodaicho, Nada-ku, Kobe 657-8501, Japan; hanswijayalipi2014@gmail.com (H.W.); akondo@kobe-u.ac.jp (A.K.)

[2] Research Center for Biotechnology, Indonesian Institute of Sciences (LIPI), Jl. Raya; Bogor Km 46, Cibinong, Bogor 16911, West Java, Indonesia; rahmani.nanik@gmail.com

[3] Graduate School of Science, Technology and Innovation, Kobe University, 1-1 Rokkodaicho, Nada-ku, Kobe 657-8501, Japan; sikengo@people.kobe-u.ac.jp (K.S.); pri@port.kobe-u.ac.jp (P.K.)

[4] Research Center for Biomaterials, Indonesian Institute of Sciences (LIPI), Jl. Raya Bogor Km 46, Cibinong, Bogor 16911, Indonesia; e_hermiati@yahoo.com

[5] National Standardization Agency of Indonesia (BSN), Gedung Badan Pengkajian dan Penerapan Teknologi (BPPT), Jl. M.H. Thamrin No. 8, Jakarta 10340, Indonesia; yopisunarya@gmail.com (Y.Y.); bambang.prasetya@gmail.com (B.P.)

[*] Correspondence: ochiaki@port.kobe-u.ac.jp; Tel./Fax: +81-78-803-6193

**Abstract:** Xylooligosaccharides (XOS) are attracting an ever-increasing amount of interest for use as food prebiotics. In this study, we used efficient membrane separation technology to convert lignocellulosic materials into a renewable source of XOS. This study revealed a dual function of nanofiltration membranes by first achieving a high yield of xylobiose (a main component of XOS) from alkali-pretreated empty fruit bunch (EFB) hydrolysate, and then by achieving a high degree of separation for xylose as a monosaccharide product. Alkali pretreatment could increase the xylan content retention of raw EFB from 23.4% to 26.9%, which eventually contributed to higher yields of both xylobiose and xylose. Nanofiltration increased the total amount of XYN10Ks_480 endoxylanase produced from recombinant *Streptomyces lividans* 1326 without altering its specific activity. Concentrated XYN10Ks_480 endoxylanase was applied to the recovery of both xylobiose and xylose from alkali-pretreated EFB hydrolysate. Xylobiose and xylose yields reached 41.1% and 17.3%, respectively, and when unconcentrated XYN10Ks_480 endoxylanase was applied, those yields reached 35.1% and 8.3%, respectively. The last step in nanofiltration was to separate xylobiose over xylose, and 41.3 g.L$^{-1}$ xylobiose (90.1% purity over xylose) was achieved. This nanofiltration method should shorten the processes used to obtain XOS as a high-value end product from lignocellulosic biomass.

**Keywords:** empty fruit bunch; xylose; xylooligosaccharides; membrane technology; endoxylanase

## 1. Introduction

The use of xylooligosaccharides (XOS) as ingredients in functional foods has increased rapidly because prebiotic oligosaccharides (OS) have various beneficial health effects, such as enhancing mineral absorption and suppressing the activity of harmful or putrefactive bacteria [1–5]. XOS are sugar oligomers comprised of xylose units through β-(1-4)-xylosidic linkages that are produced from lignocellulosic materials [1,4]. XOS are produced using chemical, enzymatic, or autohydrolysis methods.

In order to increase the production of XOS, however, the hydrolysates produced by these processes must be refined, and by-products such as monosaccharides and monosaccharide compounds must be separated to increase the purity of XOS [6–8]. For example, enzymatic hydrolysis by xylanase produces monosaccharides in addition to XOS, which lowers the concentration of XOS [9]. In our previous study, the microbial production of recombinant endo-1,4-β-xylanase (XYN10Ks_480 endoxylanase) was realized using *Streptomyces lividans* 1326 expressing an endoxylanase gene from the *Kitasatospora* sp. strain ID06-480, and the crude enzyme was free from cellulase activity [10]. This strategy enabled the co-production of xylose and XOS from acid-pretreated sugarcane bagasse [10,11]. To increase the yield of XOS, alkali pretreatment is preferable to acid pretreatment, because alkali pretreatment exhibits a minor amount of hemicellulose solubilization compared with the acid process [12]. In addition to sugarcane bagasse, another promising lignocellulosic source is empty fruit bunch (EFB), which is a waste product from palm oil plantations. Today, Indonesia and Malaysia produce approximately 85% of global crude palm oil [13]. However, the palm industry must dispose of about 1.1 tons of EFB per ton of oil produced [14]. As EFB is a non-edible part of the palm plant, it was chosen as useful lignocellulosic waste in this study [15].

The use of commercial enzymes is costly and is one of the obstacles of lignocellulose-based biorefinery technology [16]. Therefore, it is desirable to concentrate on improving total enzymatic activity. Recently, pressure-driven membrane separation processing that uses low levels of energy consumption has been used to recover enzymes and improve subsequent fermentation efficiency [17,18]. In the food industry, membrane operations are applied in the beverage industry (wine, beer, fruit juices, etc.), the dairy industry (whey protein concentration, milk protein standardization, etc.) and, to a lesser extent, in the processing of egg products [19]. In biorefinery, membrane separation has mainly been applied for biofuel recovery, sugar purification, fermentation, hydrolysis, and solvent recycling [20]. Here, the application of membrane processes to recover XOS as ingredients in functional foods from EFB were evaluated. To enhance sugar yields in the enzymatic scarification of insoluble solid fractions, one of the solutions has involved concentrating enzymes by applying membrane separation technologies. The other use of membrane separation has been to increase the purity of XOS to xylose. The purity of commercial XOS ranges from 75% to 95% [8].

The first purpose of this study was to increase the yield of XOS from alkali-pretreated EFB obtained by enzymatic hydrolysis using XYN10Ks_480 endoxylanase produced by *S. lividans* 1326, expressing an endo-xylanase gene from the *Kitasatospora* sp. strain ID06-480. This process was subjected to the nanofiltration (NF) membrane concentration process. The second purpose of this study was to investigate the purification procedure for XOS from xylose via the NF separation process. This process demonstrated the dual function of NF by improving both enzymatic yield as well as the purity of the XOS end-product.

## 2. Materials and Methods

### 2.1. Materials and Microorganisms

The EFB from *Elaeis guineensis* used in this study was obtained from Oil Palm Mill in Sukabumi, West Java, Indonesia. It was provided in the form of dry fiber. The EFB was ground using a hammer mill (Pallmann Maschinenfabrik GmbH & Co. KG, Zweibrücken, Germany), followed by a disc mill (Swan, Surabaya, Indonesia). Then, it was sieved using 40 and 60 mesh sieves to obtain a material with particle sizes equal to a 40–60 mesh and maintained in sealed plastic bags that were stored in a container.

For alkali pretreatment, 60 g of EFB was soaked in 300 mL of 3% (*w/w*) sodium hydroxide solution at room temperature for 24 h and was then filtered with Filter 70 (Strix design, Tokyo, Japan) to recover the insoluble EFB solids. The recovered solids were then washed with distilled water until pH neutrality was reached. After the solids were neutralized, they were transferred to an autoclave vessel (1 L working volume) and autoclaved at 130 °C for 8 min under 20 bar. The alkali-pretreated EFB was

stored at -20 °C until further use. This method was conducted in the same manner as in our previous report [15]. The recombinant strain used was *S. lividans* 1326 (NBRC 15675), bearing an expression system of the xylanase gene from *Kitasatospora sp.* ID06-480, as described previously [10].

## 2.2. Xylanase Production

Recombinant *S. lividans* 1326 (NBRC 15675) bearing the expression system of XYN10Ks_480 endoxylanase was inoculated into a test tube containing 5 mL of tryptic soy broth (TSB) medium (Becton, Dickinson and Company, Sparks, MD, USA) supplemented with 5 μg/mL of thiostrepton (EMD Chemicals, San Diego, CA, USA), and was then cultured at 28 °C for two days. Then, 1 mL of the seed was inoculated into a 500 mL baffled shaking flask containing 250 mL of TSB medium followed by the addition of 30 g/L glucose (Nacalai, Kyoto, Japan) as a carbon source, 15 g/L tryptone (Nacalai, Kyoto, Japan) as a nitrogen source, and 5 μg/mL of thiostrepton. To induce xylanase production, L-(-)-sorbose (Sigma-Aldrich, St. Louis, MO, USA) was added to the medium to reach a final concentration of 0.05 g/L. Cultivation was performed at 28 °C for three days.

After three days of cultivation, *S. lividans* 1326 was harvested by centrifugation at 13,000 rpm and 4 °C for 10 min and dialyzed by MEMBRA-CEL® dialysis tubing with a molecular weight cut-off (MWCO) of 3500 (RC, SERVA Electrophoresis GmbH, Heidelberg, Germany). This was followed by filtration using the following order of filter papers: polycarbonate 0.8 μm, polycarbonate 0.5 μm, and polystyrene 0.22 μm. The solution was then subjected to membrane concentration.

## 2.3. Enzymatic Activity Assay and Protein Assay

The XYN10Ks_480 endoxylanase activity was measured via a dinitrosalicylic acid (DNS) method developed by Miller with modifications [21]. Briefly, the enzymatic reaction consisted of 0.5% *w/v* beechwood xylan (Sigma-Aldrich, St. Louis, MO, USA), 50 mM sodium acetate buffer (pH 5.0), and 50 μL of the culture supernatant, incubated at 50 °C for 15 min, which was then stopped by the addition of 0.5 mL DNS solution. After being boiled for 5 min in water, the mixture was cooled on ice for 10 min and measured at 540 nm in a UV-VIS spectrophotometer (UVmini–1240, Shimadzu, Kyoto, Japan). One enzyme unit was defined as the amount of enzyme that released 1 μmol of reducing sugar for 1 min of the reaction. A protein assay was performed using a Pierce™ BCA Protein Assay Kit (Thermo Scientific™, Rockford, Illinois, USA).

## 2.4. Membrane Separation Process

RS50, a polyvinylidene fluoride ultrafiltration membrane (UF) with an MWCO of 150,000 Da, sulfonated polyethersulfone NF membrane NTR-7410 (MWCO 3000 Da), and NF membrane NTR-7450 (MWCO 600–800 Da) were obtained from the Nitto Denko Corporation (Osaka, Japan) and cut into 7.5 -cm diameter circles. The RS50 membranes were pretreated by soaking in a 50% (*v/v*) ethanol solution for 15 min, which was continued in deionized water for 15 min, with a final overnight soaking in deionized water before use. All membranes were used at 400 r/min, 25 °C, with pressures of 0.5 MPa for UF and 2.6 MPa for NF by pressure-driven nitrogen gas inside a flat membrane test cell (diameter, 104 mm; height, 147 mm; working volume, 380 mL; model, C40-B; Nitto Denko Corporation, Osaka, Japan) [22].

## 2.5. Enzymatic Hydrolysis of Pretreated EFB

The enzymatic saccharification of the pretreated EFB was performed in 50 mL screw cap Corning® centrifuge tubes (Sigma-Aldrich, Tamaulipas, Mexico). The reaction was started by adding 15% (*w/v*) of the pretreated EFB (equal to 1.8 g dry basis) into a 6 mL acetate buffer solution (pH 5) for a final volume of 50 mM, following the addition of 3.6 mL NF membrane-concentrated or unconcentrated XYN10Ks_480 endoxylanase from the *S. lividans* 1326 strain. The solution was increased to 12 mL with the addition of distilled water before incubation at 50 °C with rotation at 70 r/min. The xylose yield

was calculated using a xylose equation introduced by Pangsang, and XOS were calculated based on molecular weight [23].

## 2.6. Thin-Layer Chromatography Analysis

Thin-layer chromatography (TLC) analysis was used to identify the hydrolysis products. The hydrolysis products were filtered through a SEPARA® syringeless filter membrane (Zola Pedrosa, Italy) and detected by TLC. Sample aliquots (1 μL) were spotted four times on a TLC plate (Silica gel 60 F-254; EMD/Merck, Darmstadt, Germany), and developed in a solvent system containing *n*-butanol, acetic acid, and water (2:1.1:1, *v/v/v*). Spots were stained using DAAP reagent that contained diphenylamine, aniline, acetone, and phosphoric acid (Merck KGaA, Darmstadt, Germany), which was followed by heating at 120 °C for 15 min. The standard mixture contained xylose, xylobiose, xylotriose, xylotetraose, and xylopentaose, all of which were purchased from Megazyme (Wicklow, Ireland) [11].

## 2.7. Analytical Methods

Compositional analysis of the EFB before and after alkali pretreatment was performed using a standard procedure stipulated by the National Renewable Energy Laboratory (NREL) [24]. A high-performance liquid chromatographic (HPLC) method (Shimadzu, Kyoto, Japan) was used to determine the concentrations of xylose, xylobiose, xylotriose, xylotetraose, and xylopentaose. The standards of these compounds were obtained from Megazyme (Wicklow, Ireland). The HPLC was equipped with a refractive index (RI) detector and a column set of dual series TSKgel G2500PW$_{XL}$ (TOSOH Corporation, Tokyo, Japan) and operated at 80 °C using Milli-Q water as mobile phase at a flow rate of 0.5 mL/min.

## 3. Results and Discussion

### 3.1. Alkali Pretreatment

In this study, EFB was pretreated with alkali (sodium hydroxide) to retain hemicellulose for enzymatic hydrolysis. The chemical compositions of EFB before and after alkali pretreatment were analyzed in accordance with the NREL method [24] and appear in Table 1. A higher xylan content after pretreatment is desirable [1].

**Table 1.** Chemical composition (%, *w/w*) of empty fruit bunch (EFB) before and after alkali pretreatment.

| Sample | Xylan | Glucan | Insoluble Lignin | Soluble Lignin | Ash | Others |
|---|---|---|---|---|---|---|
| | (%) | (%) | (%) | (%) | (%) | (%) |
| Raw EFB | 23.4 ± 0.2 | 37.3 ± 0.0 | 21.2 ± 0.2 | 0.1 ± 0.0 | 0.7 ± 0.2 | 17.2 ± 0.3 |
| Alkali-pretreated EFB | 26.9 ± 1.1 | 45.0 ± 1.0 | 21.0 ± 0.4 | 0.1 ± 0.0 | 0.2 ± 0.1 | 7.5 ± 1.8 |

The alkali pretreatment increased the xylan content of raw EFB from 23.4% to 26.9%, which would eventually aid in a higher recovery of XOS [25]. The glucan content of raw EFB was also increased from 37.3% to 45.0%, which agreed with results published by Choil et al. [15]. During alkali pretreatment, the biomass amount was decreased; however, carbohydrate content increased, and lignin content was unchanged. This result corresponded with the previous report showing that the alkali pretreatment selectively removed lignin without degrading carbohydrates, thereby enhancing enzymatic hydrolysis [26].

### 3.2. Characterization of XYN10Ks_480 Endoxylanase for Membrane Selections

Alkali-pretreated EFB was enzymatically hydrolyzed by XYN10Ks_480 endoxylanase produced by recombinant *S. lividans* 1326 to obtain XOS and xylose. SDS-PAGE had previously shown that

a protein band at 49 kDa belonged to the endo-1,4-β-xylanase GH-family 10 [10]. Thus, NTR-7410 with an MWCO of 3 kDa was selected as a membrane that could be used to concentrate the xylanase [22]. The NF membrane separation process was applied to 350 mL of the culture supernatants of recombinant *S. lividans* 1326. Aliquots of culture supernatant, membrane retentate, and membrane permeate were used to measure XYN10Ks_480 endoxylanase activity and protein content.

The results of XYN10Ks_480 endoxylanase activity and protein content are listed in Table 2. Both XYN10Ks_480 endoxylanase activity and protein assay in the membrane retentate were increased compared with those in the culture supernatant. The specific activity of XYN10Ks_480 endoxylanase in the culture supernatant was similar to that in the membrane retentate. This indicated that XYN10Ks_480 endoxylanase had retained its enzyme stability even after a pressure of 2.6 MPa was applied to the system (specific activity: 6.8–6.9 U/mg). This corresponded with previous results, showing that a pressure of more than 100 MPa had slightly inactivated the xylanase activity of *Dictyoglomus thermophilum* [27].

**Table 2.** Comparison of XYN10Ks_480 endoxylanase activity and protein content before and after nanofiltration (NF) membrane separation.

| Sample | Xylanase Activity $(U.mL^{-1})$ | Protein $(g.L^{-1})$ | Specific Activity $(U.mg^{-1})$ |
|---|---|---|---|
| Recombinant *Streptomyces lividans* 1326 culture supernatant | $10.7 \pm 0.5$ | $1.5 \pm 0.0$ | $6.8 \pm 0.2$ |
| Membrane retentate | $75.3 \pm 4.1$ | $10.9 \pm 0.2$ | $6.9 \pm 0.4$ |
| Membrane permeate | 0 | $0.3 \pm 0.0$ | 0 |

*3.3. Enzymatic Hydrolysis of Alkali-Pretreated EFB with Concentrated or Unconcentrated XYN10Ks_480 Endoxylanase*

Based on the TLC results (Figure 1), xylose, xylobiose, and xylopentaose were the main products from the enzymatic hydrolysis of alkali-pretreated EFB with the use of either concentrated or unconcentrated XYN10Ks_480 endoxylanase produced from recombinant *S. lividans* 1326.

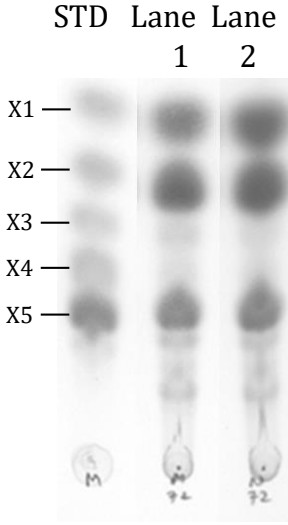

**Figure 1.** Thin-layer chromatography (TLC) results showing the primary hydrolysis products of alkali-pretreated EFB using XYN10Ks_480 endoxylanase produced from recombinant *S. lividans* 1326 in 72 h. Lane 1: Using the unconcentrated XYN10Ks_480 endoxylanase; Lane 2: Using XYN10Ks_480 endoxylanase concentrated by NF xylooligosaccharides (XOS) standards (STD); X1, xylose; X2, xylobiose; X3, xylotriose; X4, xylotetraose; X5, xylopentaose.

Figure 2 shows the XOS products by using concentrated XYN10Ks_480 endoxylanase produced from recombinant *S. lividans* 1326. The $R_f$ values of X1, X2, X3, X4, and X5 were 0.61, 0.51,

0.41, 0.33, and 0.26, respectively. Profiles were obtained after hydrolysis for 0, 24, 48, and 72 h with this recombinant enzyme (Supplementary Figure S1). As the result, X1, X2, and X5 were major XOS products by using both unconcentrated and concentrated XYN10Ks_480 endoxylanase. The hydrolysis yields were $10.0 \pm 0.4$ g.L$^{-1}$ (41.1% yield) xylobiose, $0.2 \pm 0.0$ g.L$^{-1}$ (1.2%) xylopentaose, and $7.9 \pm 1.2$ g.L$^{-1}$ (17.3%) xylose. Unconcentrated XYN10Ks_480 endoxylanase from recombinant *S. lividans* 1326, however, produced $8.6 \pm 0.7$ g.L$^{-1}$ (35.1%) xylobiose, $0.2 \pm 0.0$ g.L$^{-1}$ (1.6%) xylopentaose, and $3.8 \pm 0.4$ g.L$^{-1}$ (8.3%) xylose. Rahmani et al. [10] found that XYN10Ks_480 endoxylanase from the *Kitasatospora* sp. strain ID06-480 could produce a variety of XOS, with xylobiose as the predominant product. However, xylanase activity from the *Kitasatospora* sp. strain ID06-480 was originally weak. A further expression of endo-1,4-β-xylanase in *S. lividans* 1326 made it possible to co-produce xylobiose and xylose from alkali-pretreated EFB. This was clarified via the NF concentration of XYN10Ks_480 endoxylanase from recombinant *S. lividans* 1326. Using this NF membrane technology as a simple step can increase the total activity of the enzyme without altering the specific activity.

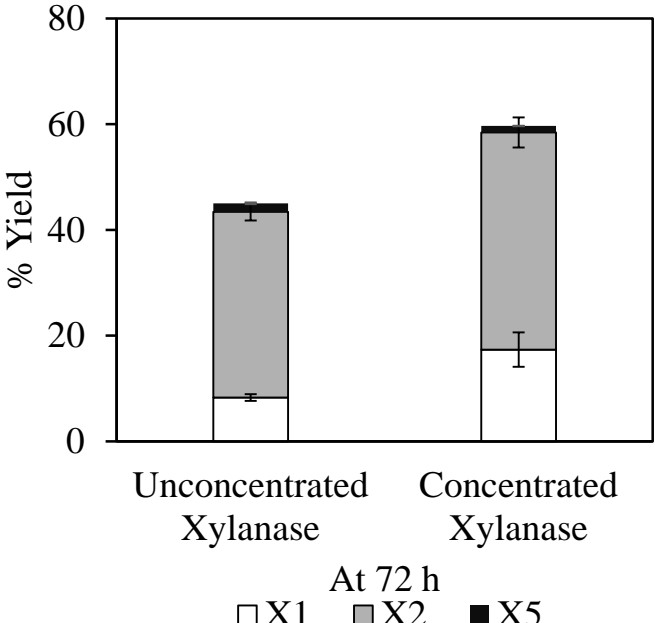

**Figure 2.** The yield from primary hydrolysis using alkali-pretreated EFB from either unconcentrated or concentrated XYN10Ks_480 endoxylanase produced by recombinant *S. lividans* 1326 in 72 h. X1, xylose; X2, xylobiose; X5, xylopentaose.

*3.4. Concentration and Separation of Xylobiose by Nanofiltration*

The primary hydrolysate from alkali-pretreated EFB using concentrated XYN10Ks_480 endoxylanase produced by recombinant *S. lividans* 1326 was subjected to sequential membrane NFs to refine and concentrate the XOS from xylose. The liquor was first subjected to an RS50 UF membrane to remove the macromolecules and to recover the sugar solution [28]. The UF permeate contained 8 g.L$^{-1}$ xylose and 11.2 g.L$^{-1}$ xylobiose (xylopentaose was a minuscule fraction, less than 1 g.L$^{-1}$). The application of NTR-7450 NF to increase xylobiose and separate xylose was repeated four times, with dilutions (up to five-fold) simultaneously administered three times, as described in Figure 3. As a result, the NTR-7450 NF membrane produced concentrated xylobiose at $41.3 \pm 6.3$ g.L$^{-1}$ for a recovery rate of 42.4% and decreased xylose to $4.1 \pm 0.3$ g.L$^{-1}$ for a recovery rate of 5.8%. The value of xylobiose against xylose achieved a maximum value of 90.1%, which compares with commercial XOS that range from 75% to 95% [8]. The NTR-7450 NF membrane, as previously described, produced concentrated sucrose while simultaneously separating glucose and fructose, and was also sufficient for the separation of xylobiose from xylose [28].

Membrane separation is a promising process for refining and concentrating XOS due to high recovery, low energy, and a simple procedure that does not require other chemicals as a solvent [6,8]. The efficient co-production of xylobiose and xylose from alkali-pretreated EFB was achieved via an integration between XYN10Ks_480 endoxylanase production from a recombinant strain and membrane concentration technology. The separation of a disaccharide (xylobiose) from a monosaccharide (xylose), was possible using the NF membrane [28]. Thus, the process proposed here could contribute to the simultaneous recovery of prebiotics and monosaccharides (the latter can be used for biochemical production using a recombinant strain) from lignocellulosic materials [8,17,18].

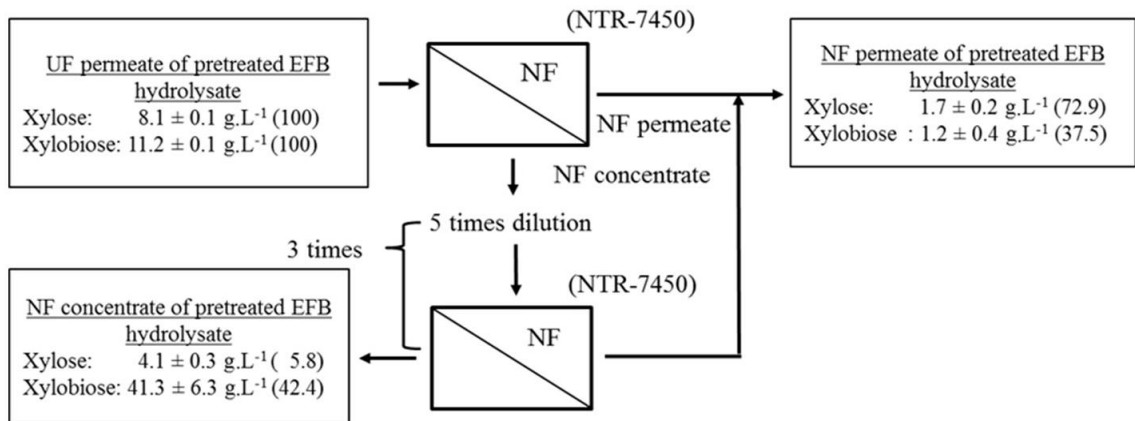

**Figure 3.** Flow chart showing the distribution of xylose and one of the xylooligosaccharides (XOS), xylobiose. Sugar recovery is shown in the parentheses. Membrane separations using ultrafiltration (UF) and NF were applied to the enzymatic hydrolysate obtained from an alkali-pretreated EFB. Data are presented as the average of the results of triplicate samples.

## 4. Conclusions

A method was successfully developed to purify XOS from a primary hydrolysate of alkali-pretreated EFB using concentrated XYN10Ks_480 endoxylanase produced by recombinant *S. lividans* 1326. An NF membrane performed the dual function of concentrating xylanase and purifying the XOS fraction. As a result, an efficient co-production of xylobiose and xylose from alkali-pretreated EFB was achieved via an integration between XYN10Ks_480 endoxylanase production from a recombinant strain and membrane separation technology. At first, alkali pretreatment retained xylan content ranging from 23.4% to 26.9% from raw EFB, which would eventually contribute to higher yields of xylobiose and xylose. The NF step was used to increase the total amount of XYN10Ks_480 endoxylanase produced from recombinant *S. lividans* 1326 without altering its specific activity. By applying the concentrated XYN10Ks_480 endoxylanase, both xylobiose and xylose from alkali-pretreated EFB hydrolysate were recovered (xylobiose and xylose yields reached 41.1% and 17.3%, respectively) in higher yields compared with the use of unconcentrated XYN10Ks_480 endoxylanase (with xylobiose and xylose yields that reached 35.1% and 8.3%, respectively). Then, NF was used to separate xylobiose over xylose, and a 41.3 g.L$^{-1}$ yield of xylobiose (90.1% purity over xylose) was achieved. Future works should focus on large-scale applications for industrial approaches.

**Supplementary Materials:** The following are available online at http://www.mdpi.com/2227-9717/8/5/619/s1. Supplementary Figure S1: Results of TLC showing the primary hydrolysis products from the enzymatic hydrolysis of alkali-pretreated EFB with either unconcentrated or concentrated XYN10Ks_480 endoxylanase produced from recombinant *S. lividans* 1326.

**Author Contributions:** H.W., K.S., and P.K. designed the research. H.W., P.K., N.R., and E.H. performed the experiments. H.W., N.R., E.H., K.S., and P.K. prepared and revised the manuscript. K.S., P.K., Y.Y., C.O., B.P., and A.K. advised on the research reported in the manuscript. All authors have read and agreed to the published version of the manuscript.

**Funding:** This research was funded by the International Joint Program, Science and Technology Research Partnership for Sustainable Development (SATREPS), Innovative BioProduction Kobe (iBioK) from the Japan Science and Technology Agency (JST), the Japan International Cooperation Agency (JICA), and The Ministry of Education, Culture, Sports, Science and Technology.

**Acknowledgments:** We appreciate the help provided by Fahriya Puspita Sari, Kharisma Panji Ramadhan, Pamela Apriliana, Ayami Fujino, and Yasunobu Takeshima.

**Conflicts of Interest:** The authors declare no conflicts of interest.

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
