# Peer review of "High Enzymatic Recovery and Purification of Xylooligosaccharides from Empty Fruit Bunch via Nanofiltration"

_processes, doi:10.3390/pr8050619_

Round 1

Reviewer 1 Report

The Authors prepared a concise and well-written manuscript on recovery and purification of xylooligosaccharides from empty fruid bench in a three-step process i.e. alkaline pre-treatment, enzymatic hydrolysis and nanofiltration. My specific comments on upgrading of the manuscript are as follows:

  1. In the introduction section, the Authors should present some more precise data on similar applications, including nanofiltration, towards lignocellulosic materials. Additionally, the novelty of the paper should be clearly stated.
  2. Materials and methods: Please provide the apparatus and related data with respect to grinding machine and spectrophotometer. Describe the grinding procedure, because it is an important and costly element of biomass pre-treatment operations.
  3. I suggest to include a bit more discussion on the alkaline pre-treatment, including process mechanism and its impact on the obtained results. Please correct a small language mistake in Tab. 1 (Alkaline or alkali not “alkai”).
  4. Please verify the quality of TLC chromatogram; for improving the discussion the Authors may consider providing the values of Rf.
  5. Conclusions: proposal of future research directions should be given.

Author Response

Response to Reviewer #1: We would like to thank the reviewer for a careful and thorough reading of this manuscript and for the thoughtful comments and constructive suggestions, which help to improve the quality of this manuscript. Our responses are follows (the reviewer’s comments are in italics).

General Comments.

The Authors prepared a concise and well-written manuscript on recovery and purification of xylooligosaccharides from empty fruid bench in a three-step process i.e. alkaline pre-treatment, enzymatic hydrolysis and nanofiltration.

Reply:

We appreciate the positive feedback from the reviewer.

With regards to improving our manuscript, as suggested by the reviewer, we have reviewed carefully the entire manuscript and have removed redundancies, as shown in the revised manuscript.

Minor comments:

  1. In the introduction section, the Authors should present some more precise data on similar applications, including nanofiltration, towards lignocellulosic materials. Additionally, the novelty of the paper should be clearly stated.

Reply:

The text has been revised as suggested:

“In biorefinery, membrane separation has been mainly applied for biofuel recovery, sugar purification, fermentation, hydrolysis, and solvent recycling [Abels et al. 2013]. Here, we tried to apply membrane process to recover XOS as ingredients in functional foods from EFB.” in P.2, line 65-68.

Adding 1 reference:

  1. Abels, C.; Carstensen, F.; Wessling, M. Membrane processes in biorefinery applications. J. Memb. Sci. 2013, 444, 285–317.

  1. Materials and methods: Please provide the apparatus and related data with respect to grinding machine and spectrophotometer. Describe the grinding procedure, because it is an important and costly element of biomass pre-treatment operations.

Reply:

The text has been revised as suggested by adding:

“at 540 nm in a UV- VIS spectrophotometer (UVmini–1240, Shimadzu, Kyoto, Japan)” in P.3, line 118.

“The EFB from Elaeis guineensis used in this study was obtained from Oil Palm Mill in Sukabumi, West Java, Indonesia. It was provided in the form of dry fiber. The EFB was ground using a hammermill (Pallmann Maschinenfabrik GmbH & Co. KG, Germany) and continued using a disc mill (Swan, Surabaya, Indonesia).  Then, it was sieved using 40 and 60 mesh sieves to obtain a material with particle sizes equal to a 40–60 mesh and maintained in sealed plastic bags that were stored in a container.” in P2., line 82-87.

  1. I suggest to include a bit more discussion on the alkaline pre-treatment, including process mechanism and its impact on the obtained results. Please correct a small language mistake in Tab. 1 (Alkaline or alkali not “alkai”).

Reply:

The text has been revised as suggested by adding:” During alkali pretreatment, biomass amount was decreased; however, carbohydrate content increased, and lignin content was unchanged. This result corresponded with the previous report shows that the alkali pretreatment selectively removes lignin without degrading carbohydrates, thereby enhancing enzymatic hydrolysis [26].”  P.4, Line 168-172.

Adding 1 reference:

  1. Kim, J.S.; Lee, Y.Y.; Kim, T.H. A review on alkaline pretreatment technology for bioconversion of lignocellulosic biomass. Bioresour. Technol. 2016, 199, 42–48.

The text has been revised as suggested by changing: ” Alkali“ in P.1, line 24; in P.4, line 161  and Table 1. “alkali” in P.2, line 88;  in P.4, line 162.

  1. Please verify the quality of TLC chromatogram; for improving the discussion the Authors may consider providing the values of Rf.

Reply:

The text has been revised as suggested by adding: ” The Rf values of X1, X2, X3, X4, and X5 were 0.61, 0.51, 0.41, 0.33 and 0.26, respectively. Profiles were obtained after hydrolysis for 0, 24, 48, and 72 h with this recombinant enzyme (Supplementary Table S1). As the result, X1, X2 and X5 were major XOS product by using both unconcentrated and concentrated XYN10Ks_480 endoxylanase. “ in P.6, line 208-211.

  1. Conclusions: proposal of future research directions should be given.

Reply:

The text has been revised as suggested by adding: ”Future works should focus on large scale applications for industrial approaches“ in P.7, line 268-269.

Reviewer 2 Report

The manuscript entitled: “High enzymatic recovery and purification of xylooligosaccharides from empty fruit bunch via nanofiltration” submitted to Processes (MDPI) as Article focuses on increasing the yield of xylooligosaccharides (XOS) using enzymatic pre-treatment with XYN10Ks_480 endoxylanase. Moreover,  the authors study nanofiltration process of purification of XOS from xylose.

The food industry is a continuously developing area supported by research, in the field of separation or enrichment of specific component. In my opinion the manuscript fits in the current trends that are of high interest for industry and academia. However, I do recommend this work to be published in Processes after a minor revision.

The list of issues than need to be addressed:

  1. Introduction: I missed about use of nanofiltration in food industry for sugar separation.
  2. Materials and Methods, 2.7 Analytical methods, page 4 of 9, lines 142-148: Could authors give more details about the exact conditions of the experiment? What was the used standard to assess the concentration and calibration curves foa quantitative assay.
  3. Results and discussion, page 4 of9,lines 151-158: Could authors clarify what was the analytical method to analyse the chemical composition of EFB from table 1.

Author Response

Response to Reviewer #2: We would like to thank the reviewer for a careful and thorough reading of this manuscript and for the thoughtful comments and constructive suggestions, which help to improve the quality of this manuscript. Our responses are follows (the reviewer’s comments are in italics).

General Comments.

The manuscript entitled: “High enzymatic recovery and purification of xylooligosaccharides from empty fruit bunch via nanofiltration” submitted to Processes (MDPI) as Article focuses on increasing the yield of xylooligosaccharides (XOS) using enzymatic pre-treatment with XYN10Ks_480 endoxylanase. Moreover,  the authors study nanofiltration process of purification of XOS from xylose.

The food industry is a continuously developing area supported by research, in the field of separation or enrichment of specific component. In my opinion the manuscript fits in the current trends that are of high interest for industry and academia. However, I do recommend this work to be published in Processes after a minor revision.

Reply:

We appreciate the positive feedback from the reviewer.

With regards to improving our manuscript, as suggested by the reviewer, we have reviewed carefully the entire manuscript and have removed redundancies, as shown in the revised manuscript.

Minor comments:

  1. Introduction: I missed about use of nanofiltration in food industry for sugar separation.

Reply:

The text has been revised as suggested:

“In the food industry, membrane operations are applied to beverage industry (wine, beer, fruit juices, etc.), the dairy industry (whey protein concentration, milk protein standardization, etc.) and, to a lesser extent, in the processing of egg products [19]. In biorefinery, membrane separation has been mainly applied for biofuel recovery, sugar purification, fermentation, hydrolysis, and solvent recycling [20]. Here, the applications of membrane process to recover XOS as ingredients in functional foods from EFB ware evaluated.” in line 63-68.

Adding 1 reference:

  1. Echavarría, A.P.; Torras, C.; Pagán, J.; Ibarz, A. Fruit Juice Processing and Membrane Technology Application. Food Eng. Rev. 2011, 3, 136–158.
  2. Abels, C.; Carstensen, F.; Wessling, M. Membrane processes in biorefinery applications. J. Memb. Sci. 2013, 444, 285–317.

  1. Materials and Methods, 2.7 Analytical methods, page 4 of 9, lines 142-148: Could authors give more details about the exact conditions of the experiment? What was the used standard to assess the concentration and calibration curves foa quantitative assay.

Reply:

The text has been revised as suggested by adding: “The standards of these compounds were obtained from Megazyme (Wicklow, Ireland). “ in P.4, line 155-156.

  1. Results and discussion, page 4 of9,lines 151-158: Could authors clarify what was the analytical method to analyse the chemical composition of EFB from table 1.

Reply:

The text has been revised as suggested by adding: ” The chemical compositions of EFB before and after alkali pretreatment were analyzed in accordance with NREL method [28] and appear in Table 1.” in P.4, line 163-164.
